# A Schmidt Decomposition Approach to Quantum Thermodynamics

**DOI:** 10.3390/e24111645

**Published:** 2022-11-12

**Authors:** André Hernandes Alves Malavazi, Frederico Brito

**Affiliations:** Instituto de Física de São Carlos, Universidade de São Paulo, C.P. 369, São Carlos 13560-970, SP, Brazil

**Keywords:** quantum thermodynamics, open quantum systems, strong coupling

## Abstract

The development of a self-consistent thermodynamic theory of quantum systems is of fundamental importance for modern physics. Still, despite its essential role in quantum science and technology, there is no unifying formalism for characterizing the thermodynamics within general autonomous quantum systems, and many fundamental open questions remain unanswered. Along these lines, most current efforts and approaches restrict the analysis to particular scenarios of approximative descriptions and semi-classical regimes. Here, we propose a novel approach to describe the thermodynamics of arbitrary bipartite autonomous quantum systems based on the well-known Schmidt decomposition. This formalism provides a simple, exact, and symmetrical framework for expressing the energetics between interacting systems, including scenarios beyond the standard description regimes, such as strong coupling. We show that this procedure allows straightforward identification of local effective operators suitable for characterizing the physical local internal energies. We also demonstrate that these quantities naturally satisfy the usual thermodynamic notion of energy additivity.

## 1. Introduction

In previous decades, there have been many efforts to extend thermodynamics results and concepts to both encompass microscopic systems out of equilibrium [1,2] and to include genuine quantum phenomena. This context, aligned with the state-of-the-art capability to manipulate delicate quantum systems across a wide variety of physical platforms, has been fomenting the current endeavours to develop a self-consistent thermodynamic theory of quantum systems [3,4,5,6]. Interestingly, despite all recent extensive research and advances in quantum thermodynamics [7,8,9,10,11,12,13], some open questions are exceptionally notorious, especially those concerning central aspects of the theory. In particular, there are no universally accepted general definitions for the quantum counterparts of the most basic thermodynamic quantities and, therefore, no formalism has been consolidated for a quantum thermodynamic description suitable for arbitrary quantum systems. This context poses a fundamental restriction to many cases of interest and highlights a core issue in the field.

While there is no ambiguity in defining proper thermodynamic quantities for macroscopic systems in equilibrium, fundamental subtleties arise once one must account for non-negligible interactions and intrinsically quantum properties of out-of-equilibrium systems. In this sense, the identification of physical, local, internal energy and quantum thermodynamic entropy is not as straightforward as in the classical scenario, since it is unclear how to proceed in the presence of these notions. Although different definition approaches have been reported in the literature [14,15,16,17,18,19,20,21], this task remains elusive and far from consensus. Naturally, any attempt to describe energy exchanges, such as work and heat, inherits this fundamental uncertainty and brings its own conceptual issues and challenges [22,23,24,25,26,27].

Additionally, most current approaches in quantum thermodynamics commonly rely on a particular class of assumptions and approximations, which inevitably limits their application to a small set of regimes. In particular, it is often assumed weakly-coupled systems, Markovian dynamics, and semi-classical external control are present [14,28,29,30]. Despite being well-suited for specific scenarios, such assumptions clearly exclude several relevant situations.

Along these lines, it is clear that the current status of the theory is restrictive and does not allow for the description and characterization of thermodynamic processes for general contexts, especially for autonomous quantum systems and in the presence of strong coupling. In this respect, many efforts have been made to expand quantum thermodynamics to these broader conditions [15,16,18,31,32,33,34,35,36,37]. Nevertheless, there is no general framework that is suitable for arbitrary quantum systems.

This work is inserted in this context. Here, we introduce a novel approach for analyzing the energetic exchange occurring between parts of a generic autonomous quantum system. The proposal presented here is exact and follows the notorious mathematical procedure of the Schmidt decomposition for bipartite quantum systems. Interestingly, despite its simplicity and powerful statement, it is still not explicitly and widely considered in the context of quantum thermodynamics. Such a framework will permit us to describe the dynamics and energetics within general interacting subsystems in a symmetrical manner, i.e., regardless of their individual local properties, specific details, and dimensionality, they will be treated equally. Moreover, it will not demand any approximations and complementary hypotheses, such as the commonly used ones concerning the interaction regimes, characteristics of the dynamics, and Hamiltonian structure, i.e., weak-coupling, Markovianity, and strict energy conservation. From a formal perspective, we introduce time-dependent local effective Hamiltonians for the subsystems naturally containing their respective bare ones and the contributions of the interaction term. We identify these elements as the representative operators to characterize the subsystem’s physical internal energies, which allows us to extend the valuable classical thermodynamic notion of energy additivity to arbitrary interacting bipartite quantum systems. Such identification represents one of our main results.

## 2. Results

### 2.1. System Description

As a quantum universe, we consider a finite and isolated quantum system composed of two interacting subsystems with dimensions $d^{\left(1\right)}$ and $d^{\left(2\right)}$. In addition, let us assume that—without any loss of generality—$d^{\left(1\right)}\leq d^{\left(2\right)}$. The whole system Hamiltonian reads
(1)$${\hat{H}}^{\left(0\right)}:={\hat{H}}^{\left(1\right)}⊗{\hat{1}}^{\left(2\right)}+{\hat{1}}^{\left(1\right)}⊗{\hat{H}}^{\left(2\right)}+{\hat{H}}_{int},$$
where ${\hat{H}}^{\left(k\right)}$ is the local bare Hamiltonian of subsystem (k) with k=1,2 and ${\hat{H}}_{int}$ is the term encompassing all interactions between them.

At any time *t*, the universe’s quantum state is depicted by a pure state represented by the state vector |Ψ(t)〉, whose dynamics are unitary and ditacted by the Schrödinger equation $i\hbar \frac{d}{dt}|\Psi \left(t\right)\rangle ={\hat{H}}^{\left(0\right)}|\Psi \left(t\right)\rangle$. The local state of each subsystem is determined by the reduced density matrix of the universe, such that ${\hat{\rho }}^{\left(1,2\right)}\left(t\right)\equiv tr_{2,1}\{|\Psi \left(t\right)\rangle \langle \Psi \left(t\right)|\}$. No additional assumptions regarding the universe, its constituents, or the Hamiltonians are considered.

### 2.2. Schmidt Decomposition Approach

Naturally, |Ψ(t)〉 can be written according to any conceivable basis. Nevertheless, the well known Schmidt decomposition [38] provides the following particular and convenient form
(2)$$|\Psi \left(t\right)\rangle =\sum_{j=1}^{d^{\left(1\right)}}\lambda_{j}\left(t\right)|\varphi_{j}^{\left(1\right)}\left(t\right)\rangle ⊗|\varphi_{j}^{\left(2\right)}\left(t\right)\rangle ,$$
for every time *t*, where $\left\{\lambda_{j}\left(t\right)\geq 0;j=1,\ldots ,d^{\left(1\right)}\right\}$ and $\{|\varphi_{j}^{\left(k\right)}\left(t\right)\rangle ;j=1,\ldots ,d^{\left(1\right)}\}$ are the time-local Schmidt coefficients and local Schmidt basis of subsystem (k), respectively. Considering its form, the representation given by Equation (Equation 2) will prove itself a handful and compelling one for a number of reasons. Notice that, despite a potential distinction between $d^{\left(1\right)}$ and $d^{\left(2\right)}$, the vector state decomposition runs in a single-index sum that is bounded by the smallest dimension in question, the hypothesis $d^{\left(1\right)}$. Furthermore, there is a one-to-one relation between the Schmidt basis elements, i.e., each Schmidt ket $|\varphi_{j}^{\left(1\right)}\left(t\right)\rangle$ from subsystem (1) is related to subsystem (2) Schmidt ket $|\varphi_{j}^{\left(2\right)}\left(t\right)\rangle$. Additionally, it is guaranteed that the Schmidt coefficients $\lambda_{j}\left(t\right)$ are unambiguously defined, while the Schmidt bases are unique up to eventual degenerate coefficients and a phase degree of freedom, in the sense that Equation (Equation 2) is invariant over simultaneous local phases changes. Additionally, it already provides the pure-orthonormal ensemble representation of the subsystem’s local density matrices ${\hat{\rho }}^{\left(k\right)}\left(t\right)$, i.e., their simultaneous spectral decompositions are given by the squared Schmidt coefficients ${\left\{\lambda_{j}^{2}\left(t\right)\right\}}_{j}$ and the Schmidt basis $\{|\varphi_{j}^{\left(k\right)}\left(t\right)\rangle {\}}_{j}$, such that ${\hat{\rho }}^{\left(k\right)}\left(t\right)=\sum_{j=1}^{d^{\left(1\right)}}\lambda_{j}^{2}\left(t\right)|\varphi_{j}^{\left(k\right)}\left(t\right)\rangle \langle \varphi_{j}^{\left(k\right)}\left(t\right)|$. In particular, notice that both local state representations are—in general—of mixed state and have the same spectrum whenever the universe is pure.

The autonomous time evolution executed by the whole system at any time interval $\left[t_{0},t_{1}\right]$ also induces the local dynamic changes, which, in turn, are mainly non-unitary and strongly influenced by the presence of the interaction term ${\hat{H}}_{int}$. Let us now focus on the changes in the Schmidt decomposition elements and define the local dynamical maps ${\tilde{U}}^{\left(k\right)}:H^{\left(k\right)}\to H^{\left(k\right)}$ as the operators characterizing the Schmidt basis dynamics, i.e., $|\varphi_{j}^{\left(k\right)}\left(t\right)\rangle ={\tilde{U}}^{\left(k\right)}\left(t,t_{0}\right)|\varphi_{j}^{\left(k\right)}\left(t_{0}\right)\rangle$ for any $t\geq t_{0}$ with $\lim_{t\to t_{0}}\;{\tilde{U}}^{\left(k\right)}\left(t,t_{0}\right)={\hat{1}}^{\left(k\right)}$. Since the Schmidt basis at distinct times corresponds to different orthonormal bases for the same Hilbert space, the relationship above is unitary, and one can define the local time-translation generators ${\tilde{H}}^{\left(k\right)}\left(t\right)={\tilde{H}}^{(k)†}\left(t\right)$ of the Schmidt basis $\{|\varphi_{j}^{\left(k\right)}\left(t\right)\rangle {\}}_{j}$, such that $i\hbar \frac{d}{dt}{\tilde{U}}^{\left(k\right)}\left(t,t_{0}\right)={\tilde{H}}^{\left(k\right)}\left(t\right){\tilde{U}}^{\left(k\right)}\left(t,t_{0}\right)$, and therefore,
(3)$$i\hbar \frac{d}{dt}|\varphi_{j}^{\left(k\right)}\left(t\right)\rangle ={\tilde{H}}^{\left(k\right)}\left(t\right)|\varphi_{j}^{\left(k\right)}\left(t\right)\rangle ,$$
for all *j*. Alternatively, ${\tilde{H}}^{\left(k\right)}\left(t\right)$ can be simply cast as
(4)$${\tilde{H}}^{\left(k\right)}\left(t\right)\equiv i\hbar \sum_{j=1}^{d^{\left(k\right)}}\left(\frac{d}{dt}|\varphi_{j}^{\left(k\right)}\left(t\right)\rangle \right)\langle \varphi_{j}^{\left(k\right)}\left(t\right)|.$$
Still, we can go one step further and show that Equation (Equation 4) above can be directly related to the local bare Hamiltonian ${\hat{H}}^{\left(k\right)}$, as follows (see Section 4)
(5)$${\tilde{H}}^{\left(k\right)}\left(t\right)={\hat{H}}^{\left(k\right)}+{\hat{H}}_{LS}^{\left(k\right)}\left(t\right)+{\hat{H}}_{X}^{\left(k\right)}\left(t\right),$$
i.e., ${\tilde{H}}^{\left(k\right)}\left(t\right)$ can be split into the sum of three distinct elements, including the bare Hamiltonian. The additional operators are responsible for the time dependency of ${\tilde{H}}^{\left(k\right)}\left(t\right)$, where ${\hat{H}}_{LS}^{\left(k\right)}\left(t\right)$ is a general Lamb-shift-like term, in the sense that it is diagonal in the bare eigenbasis, i.e., $[{\hat{H}}_{LS}^{\left(k\right)}\left(t\right),{\hat{H}}^{\left(k\right)}]=0$ for all *t*, and ${\hat{H}}_{X}^{\left(k\right)}\left(t\right)$ contains only non-diagonal elements. Note that, in the absence of interactions between the partitions, both subsystems will individually evolve over time in a unitary fashion. Under these circumstances, the role of the time-translation generator of the Schmidt basis will be naturally assigned to the respective local bare Hamiltonians. That is, the free evolution of the Schmidt basis will be simply $|{\bar{\varphi }}_{j}^{\left(k\right)}\left(t\right)\rangle =e^{-\frac{i}{\hbar }{\hat{H}}^{\left(k\right)}\left(t-t_{0}\right)}|\varphi_{j}^{\left(k\right)}\left(t_{0}\right)\rangle$. In this sense, it is clear that (i) the elements ${\hat{H}}_{LS}^{\left(k\right)}\left(t\right)$ and ${\hat{H}}_{X}^{\left(k\right)}\left(t\right)$ are direct local byproducts of the interaction term and vanish for non-interacting systems; (ii) Equation (Equation 5) acts as an effective Hamiltonian. Thus, from now on, ${\tilde{H}}^{\left(k\right)}\left(t\right)$ will be referred to as the *local effective Hamiltonian* of subsystem (k).

Additionally, the local effective Hamiltonians are responsible for guiding the unitary component of the dynamics of ${\hat{\rho }}^{\left(k\right)}\left(t\right)$, i.e.,
(6)$$i\hbar \frac{d}{dt}{\hat{\rho }}^{\left(k\right)}\left(t\right)=\left[{\tilde{H}}^{\left(k\right)}\left(t\right),{\hat{\rho }}^{\left(k\right)}\left(t\right)\right]+{\hat{D}}_{t}^{\left(k\right)}{\hat{\rho }}^{\left(k\right)}\left(t\right),$$
where ${\hat{D}}_{t}^{\left(k\right)}{\hat{\rho }}^{\left(k\right)}\left(t\right)=i\hbar \sum_{j=1}^{d^{\left(1\right)}}\left(\frac{d}{dt}\lambda_{j}^{2}\left(t\right)\right)|\varphi_{j}^{\left(k\right)}\left(t\right)\rangle \langle \varphi_{j}^{\left(k\right)}\left(t\right)|$ is the non-unitary part that explicitly depends on the time-functional form of the Schmidt coefficients.

### 2.3. Internal Energies and Additivity

By construction, in the Schrödinger picture, the time-translation generator of isolated and closed quantum systems *is* the functional of energy. Thus, the total internal energy, $U^{\left(0\right)}$, contained within the universe under consideration is completely specified by the bare Hamiltonian ${\hat{H}}^{\left(0\right)}$, i.e., one can readily identify
$$U^{\left(0\right)}\equiv \langle {\hat{H}}^{\left(0\right)}\rangle$$
where 〈.〉≡〈Ψ(t)|(.)|Ψ(t)〉. Additionally, since the bipartition is isolated, no energy flows inward or outward of the system. Along these lines, it is easy to see that the unitary evolution of |Ψ(t)〉 naturally guarantees internal energy conservation, such that
(7)$$\frac{d}{dt}U^{\left(0\right)}=0.$$
Interestingly, it is unclear how to obtain a consistent and meaningful analogy for the interacting subsystems. In general, the notion of local internal energy is blurred by non-negligible interactions within the whole system, i.e., the total internal energy can be written as the sum of the individual local contributions of the expectation values of the bare Hamiltonians and the interaction term,
$$U^{\left(0\right)}=\langle {\hat{H}}^{\left(1\right)}\rangle \left(t\right)+\langle {\hat{H}}^{\left(2\right)}\rangle \left(t\right)+\langle {\hat{H}}_{int}\rangle \left(t\right).$$
Thus, unless considering particular scenarios, it is clear that the interaction is indispensable for the computation of the universe’s energy. Yet, its role is uncertain when one attempts to establish the local observable to characterize the subsystem’s internal energy. Along these lines, given that the interaction term actively influences their local dynamics, one could expect that its contribution should be shared between them in one way or another.

In this context, it is desirable that two relevant properties for any suitable definition of local internal energies (i) are determinable by local measurements, i.e., associated with the expectation value of a local operator and (ii) are an additive quantity. While the first condition guarantees the means for a local characterization, the second also allows for the intuitive picture of energy flowing from one system to another without including energetic sinks or sources, i.e., the sum of the local internal energies is a conserved quantity. These features, of course, are not trivial, especially because the interaction term is a global feature. This fact, however, suggests that an effective approach for describing local internal energy provides the most promising route. Otherwise, a global picture would be necessary for characterizing the energetic flux, which is impractical for most realistic scenarios.

We first remark that the expectation value of the local effective Hamiltonians presented earlier satisfies both desired features for being interpreted as the representative quantities for characterizing the subsystem’s physical internal energies. Thus, by identifying
(8)$$U^{\left(k\right)}\left(t\right):=\langle {\tilde{H}}^{\left(k\right)}\left(t\right)\rangle ,$$
the local internal energies become locally accessible properties, and one can show that their sum is equal to the constant universe’s internal energy, such that
(9)$$U^{\left(0\right)}=U^{\left(1\right)}\left(t\right)+U^{\left(2\right)}\left(t\right)$$
and all energy flowing from subsystem (1) is fully transferred to subsystem (2), i.e., no additional energetic source is required (see Section 4). Additionally, by directly relating these general expressions to the universe’s internal energy, the relationship between the operators ${\hat{H}}_{LS}^{\left(k\right)}\left(t\right)$, ${\hat{H}}_{X}^{\left(k\right)}\left(t\right)$ and the interaction term becomes more apparent. Indeed, the expectation value $\langle {\hat{H}}_{int}\rangle \left(t\right)$ is such that one finds $\langle {\hat{H}}_{int}\rangle \left(t\right)=\langle {\hat{H}}_{LS}^{\left(1\right)}\left(t\right)\rangle +\langle {\hat{H}}_{X}^{\left(1\right)}\left(t\right)\rangle +\langle {\hat{H}}_{LS}^{\left(2\right)}\left(t\right)\rangle +\langle {\hat{H}}_{X}^{\left(2\right)}\left(t\right)\rangle$. It is worth mentioning that their numerical values might be extremely different, despite the symmetrical decomposition form seen above.

Note that such a description is general, symmetrical, and does not rely on approximation, particular regimes, or additional restrictive hypotheses concerning the subsystem dynamics, the Hamiltonian structure, and the coupling strength. This identification represents one of our main results.

### 2.4. Phase Gauge

As mentioned earlier, the Schmidt basis is unique up to the eventual degenerate Schmidt coefficients and a phase choice freedom. While the former is unimportant to our current purposes, the latter brings nontrivial consequences to our formalism. From a global perspective, it is easy to see that the simultaneous addition of local phases ${\left\{\theta_{j}\left(t\right)\right\}}_{j}\in R$, such that
(10)$$|\varphi_{j}^{\prime (k)}\left(t\right)\rangle =e^{{\left(-1\right)}^{k-1}i\theta_{j}\left(t\right)}|\varphi_{j}^{\left(k\right)}\left(t\right)\rangle ,$$
maintains the universe state form in an unchanged state, i.e., the phases cancel out and $|\Psi^{\prime }\left(t\right)\rangle =|\Psi \left(t\right)\rangle$. Such phase invariance expresses an internal gauge freedom choice inbuilt in the Schmidt decomposition that directly affects the identification of the local effective Hamiltonians, i.e., instead of describing the state vector with the basis $\{|\varphi_{j}^{\left(k\right)}\left(t\right)\rangle {\}}_{j}$, one can use $\{|\varphi_{j}^{\prime (k)}\left(t\right)\rangle {\}}_{j}$ and find its time-translation generator ${\tilde{H}}^{\prime (k)}\left(t\right)$. Given Equation (Equation 4), the gauge transformation defined in Equation (Equation 10) above implies the following expression for ${\tilde{H}}^{\prime (k)}\left(t\right)$ in terms of ${\tilde{H}}^{\left(k\right)}\left(t\right)$ and ${\left\{\theta_{j}\left(t\right)\right\}}_{j}$
(11)$${\tilde{H}}^{\prime (k)}\left(t\right)={\tilde{H}}^{\left(k\right)}\left(t\right)+{\left(-1\right)}^{k}\hbar \sum_{j=1}^{d^{\left(k\right)}}\left(\frac{d\theta_{j}\left(t\right)}{dt}\right)|\varphi_{j}^{\left(k\right)}\left(t\right)\rangle \langle \varphi_{j}^{\left(k\right)}\left(t\right)|.$$
Naturally, its expectation value also transforms accordingly, such that
(12)$$\langle {\tilde{H}}^{\prime (k)}\left(t\right)\rangle =\langle {\tilde{H}}^{\left(k\right)}\left(t\right)\rangle +{\left(-1\right)}^{k}\hbar \sum_{j=1}^{d^{\left(k\right)}}\lambda_{j}^{2}\left(t\right)\left(\frac{d\theta_{j}\left(t\right)}{dt}\right).$$
Thus, it is clear that from a local point of view that these phases are relevant and should be carefully scrutinized.

Note that a gauge change adds an extra diagonal term on the Schmidt basis that only depends on the time derivative of the phases. In general, these additional quantities change the local effective Hamiltonian structure such that both their eigenbasis and eigenvalues are affected, which also implies that the spectral gaps are not necessarily invariant. Still, as expected, such transformations maintain the description of the local density matrices and their dynamical Equation (Equation 6) in an unchanged state, given that $[|\varphi_{j}^{\left(k\right)}\left(t\right)\rangle \langle \varphi_{j}^{\left(k\right)}\left(t\right)|,{\hat{\rho }}^{\left(k\right)}\left(t\right)]=0$ for all *j* and *t*. More importantly, these terms shift the expectation values $\langle {\tilde{H}}^{\left(k\right)}\left(t\right)\rangle$ in such a way that the change obtained by subsystem (1) is compensated by the one acquired by subsystem (2). Hence, both the universe’s internal energy $U^{\left(0\right)}$ and the additivity property are guaranteed to be invariant over these modifications, i.e., $U^{\prime (0)}=U^{\left(0\right)}$ and $\langle {\tilde{H}}^{\prime (1)}\left(t\right)\rangle +\langle {\tilde{H}}^{\prime (2)}\left(t\right)\rangle =\langle {\tilde{H}}^{\left(1\right)}\left(t\right)\rangle +\langle {\tilde{H}}^{\left(2\right)}\left(t\right)\rangle$.

Hence, given the freedom of choosing the Schmidt basis for writing the state vector, it is clear that, for the same bipartite quantum system, several local effective Hamiltonians are suitable for consistently describing the subsystem’s dynamics, any observable physical quantity, and the universe energetics. However, despite ${\tilde{H}}^{\left(k\right)}\left(t\right)$ possessing the desired properties to successfully fulfill the role of physical local internal energies, such freedom also brings a significant ambiguity in identifying a single pair of preferential operators and quantifying them.

Notably, such mathematical freedom of adding local phases is not necessarily physically consistent for characterizing energy in general when seeking its association with the time-translation generator of the dynamics [39]. As mentioned earlier, if there are no interactions between the bipartitions, they will behave independently as isolated quantum systems. In this scenario, the Schmidt basis dynamics can be described by $i\hbar \frac{d}{dt}|{\bar{\varphi }}_{j}^{\left(k\right)}\left(t\right)\rangle ={\hat{H}}^{\left(k\right)}|{\bar{\varphi }}_{j}^{\left(k\right)}\left(t\right)\rangle$, and the individual local internal energies can be readily identified as $U^{\left(k\right)}\equiv \langle {\hat{H}}^{\left(k\right)}\rangle$, where $\frac{d}{dt}U^{\left(k\right)}=0$. Along these lines, once considering the interaction term negligible, similar conclusions should be obtained regardless of the chosen gauge. In this respect, if $\{|\varphi_{j}^{\left(k\right)}\left(t\right)\rangle {\}}_{j}$ is indeed the Schmidt basis, such that $|\varphi_{j}^{\left(k\right)}\left(t\right)\rangle \to |{\bar{\varphi }}_{j}^{\left(k\right)}\left(t\right)\rangle$ when ${\hat{H}}_{int}\to 0$, then it is guaranteed that ${\tilde{H}}^{\left(k\right)}\left(t\right)\to {\hat{H}}^{\left(k\right)}$ and, since the phases are arbitrary and independent of the interaction term, ${\tilde{H}}^{\prime (k)}\left(t\right)\to {\hat{H}}^{\left(k\right)}+{\left(-1\right)}^{k}\hbar \sum_{j=1}^{d^{\left(k\right)}}\left(\frac{d\theta_{j}\left(t\right)}{dt}\right)|{\bar{\varphi }}_{j}^{\left(k\right)}\left(t\right)\rangle \langle {\bar{\varphi }}_{j}^{\left(k\right)}\left(t\right)|$. Notice that, while the former is the time-translation generator of the basis $\{|{\bar{\varphi }}_{j}^{\left(k\right)}\left(t\right)\rangle {\}}_{j}$, the latter is the one relative to the basis $\{e^{{\left(-1\right)}^{k-1}i\theta_{j}\left(t\right)}|{\bar{\varphi }}_{j}^{\left(k\right)}\left(t\right)\rangle {\}}_{j}$. More importantly, it is clear that, in general, not all local effective Hamiltonians (phases) are suitable for correctly describing local energy measurements in the limit behavior of non-interacting systems. Nevertheless, this is achieved iff $\frac{d}{dt}\theta_{j}\left(t\right)=\alpha$ for all *j*, since additive constants only shift the bare Hamiltonian spectrum. Thus, as long all phases are equal linear functions of time, we have a physically consistent set of gauges satisfying ${\tilde{H}}^{\prime (k)}\left(t\right)={\tilde{H}}^{\left(k\right)}\left(t\right)+{\left(-1\right)}^{k}\hbar \alpha {\hat{1}}^{\left(k\right)}$ and $\langle {\tilde{H}}^{\prime (k)}\left(t\right)\rangle =\langle {\tilde{H}}^{\left(k\right)}\left(t\right)\rangle +{\left(-1\right)}^{k}\hbar \alpha$ in a way that the remaining freedom is also guaranteed to maintain the spectral gaps of the local effective Hamiltonian invariant. Despite being local operators, observe that such a procedure requires knowledge of the “whole” in order to determine the “parts” energetics, i.e., obtaining the physical local effective Hamiltonians requires knowledge about the interaction of the parts (a non-local property) to fix the correct physical phase gauge. Hence, in order to construct a local and consistent energy description for the parts, one cannot rely solely on local features (see Reference [40] for a recent discussion).

In summary, by restricting ourselves to gauges such that ${\left\{\frac{d}{dt}\theta_{j}\left(t\right)=\alpha \right\}}_{j}$, where α∈R, one can identify the expectation values $\langle {\tilde{H}}^{\left(k\right)}\left(t\right)\rangle$ as the local physical internal energies up to an additive constant, $U^{\left(k\right)}\left(t\right):=\langle {\tilde{H}}^{\left(k\right)}\left(t\right)\rangle$ and $U^{\prime (k)}\left(t\right)=U^{\left(k\right)}\left(t\right)+{\left(-1\right)}^{k}\hbar \alpha$. In this sense, even though different gauges provide distinct absolute internal energy values, energy measurements differences remain identical.

## 3. Conclusions

In this work, we introduced a novel formalism suitable for describing the energetics within arbitrary autonomous quantum systems. Despite being presented for pure quantum states, the generalization for mixed ones is straightforward (see Section 4). Formally, the procedure is based on the well-known Schmidt decomposition given by Equation (Equation 2), which provides the basis for the identification of the local effective Hamiltonians defined by Equation (Equation 4). We highlighted the fact that these operators possess the desired properties for being considered suitable candidates for characterizing the subsystem’s internal energies, i.e., their expectation values are local quantities and naturally satisfy the classical thermodynamic notion of energy additivity for the bipartition, as shown in Equation (Equation 9). In contrast with current approaches, such a framework is exact, treats both partitions on equal footing, and does not rely on additional restrictive hypotheses and approximations, i.e., specific types of dynamics, convenient Hamiltonian structures, particular coupling regimes and semi-classical descriptions.

Along with such an identification, we verified the existence of an internal phase gauge freedom corresponding to the phase choices ${\left\{\theta_{j}\left(t\right)\right\}}_{j}$ within the Schmidt decomposition procedure. As shown in Equations (Equation 11) and (Equation 12), this arbitrary choice affects the structure of the local effective Hamiltonians and, in general, manifests as a time-dependent contribution to their expectation values. Nevertheless, despite the mathematical freedom, physical consistency during gauge transformations is only achieved for phases such that $\frac{d}{dt}\theta_{j}\left(t\right)=\alpha$ for all *j*, where α∈R. In this case, we have ${\tilde{H}}^{\prime (k)}\left(t\right)={\tilde{H}}^{\left(k\right)}\left(t\right)+{\left(-1\right)}^{k}\hbar \alpha {\hat{1}}^{\left(k\right)}$ and $\langle {\tilde{H}}^{\prime (k)}\left(t\right)\rangle =\langle {\tilde{H}}^{\left(k\right)}\left(t\right)\rangle +{\left(-1\right)}^{k}\hbar \alpha$, and the limit behavior ${\tilde{H}}^{\prime (k)}\left(t\right)\to {\hat{H}}^{\left(k\right)}+{\left(-1\right)}^{k}\hbar \alpha {\hat{1}}^{\left(k\right)}$ is satisfied for ${\hat{H}}_{int}\to 0$. Thus, by identifying the local internal energies as $U^{\left(k\right)}\left(t\right):=\langle {\tilde{H}}^{\left(k\right)}\left(t\right)\rangle$, different gauges would provide distinct absolute values for these quantities but equivalent spectral gaps. In this sense, the energy shift ${\left(-1\right)}^{k}\hbar \alpha$ is analogous to the classical thermodynamic freedom in the definition of internal energy [41].

In the current status of quantum thermodynamics, where there is no general methodology suitable for describing the energetic exchanges of arbitrary interacting quantum systems on equal footing, the procedure devised here represents a viable alternative to fill this important gap and a promising route for the characterization of general quantum thermodynamic processes. In particular, it provides a means to describe scenarios that do not fall under the standard description regimes, especially for strongly coupled systems, and in the absence of external classical agents. Additionally, the presented formalism has the potential to aid in the understanding and characterization of the thermodynamic role played by entanglement, especially during these energy exchanges. This understanding is also of fundamental interest for practical purposes, such as designing genuine quantum heat engines. One may ask about the eventual energetic differences and similarities between bound and free entangled states [42,43] (see [44] for a recent contribution to this question and [45] for the relationship between the Schmidt number and bound entanglement). Investigations on this matter establish an interesting and relevant avenue for future work.

## 4. Additional Information

### 4.1. Local Effective Hamiltonian

Given the following spectral decomposition for the local bare Hamiltonian
(13)$${\hat{H}}^{\left(k\right)}\equiv \sum_{j=1}^{d^{\left(k\right)}}b_{j}^{\left(k\right)}|b_{j}^{\left(k\right)}\rangle \langle b_{j}^{\left(k\right)}|,$$
where ${\left\{b_{j}^{\left(k\right)}\right\}}_{j}$ and $\{|b_{j}^{\left(k\right)}\rangle {\}}_{j}$ are its respective bare eigenenergies and eigenbasis, one can define the projections $\langle b_{j}^{\left(k\right)}|\varphi_{l}^{\left(k\right)}\left(t\right)\rangle :=r_{jl}^{\left(k\right)}\left(t\right)e^{-\frac{i}{\hbar }b_{j}^{\left(k\right)}t}$, such that
$$\langle b_{j}^{\left(k\right)}|\frac{d}{dt}|\varphi_{l}^{\left(k\right)}\left(t\right)\rangle =\left(\frac{d}{dt}r_{jl}^{\left(k\right)}\left(t\right)\right)e^{-\frac{i}{\hbar }b_{j}^{\left(k\right)}t}-\frac{i}{\hbar }b_{j}^{\left(k\right)}r_{jl}^{\left(k\right)}\left(t\right)e^{-\frac{i}{\hbar }b_{j}^{\left(k\right)}t}.$$
Additionally, given the orthonormality $\langle b_{\alpha }^{\left(k\right)}|b_{\beta }^{\left(k\right)}\rangle =\delta_{\alpha \beta }$, it is easy to see that
$$\sum_{l=1}^{d^{\left(k\right)}}r_{\alpha l}^{\left(k\right)}\left(t\right){\left(r_{\beta l}^{\left(k\right)}\left(t\right)\right)}^{*}e^{\frac{i}{\hbar }\left(b_{\beta }^{\left(k\right)}-b_{\alpha }^{\left(k\right)}\right)t}=\delta_{\alpha \beta }.$$
Finally, by casting ${\tilde{H}}^{\left(k\right)}\left(t\right)$ in the bare eigenbasis representation and using the previous relations, one can rewrite the local effective Hamiltonian according to Equation (Equation 5), i.e.,
$${\tilde{H}}^{\left(k\right)}\left(t\right)={\hat{H}}^{\left(k\right)}+{\hat{H}}_{LS}^{\left(k\right)}\left(t\right)+{\hat{H}}_{X}^{\left(k\right)}\left(t\right),$$
where
$${\hat{H}}_{LS}^{\left(k\right)}\left(t\right)\equiv i\hbar \sum_{j=1}^{d^{\left(k\right)}}\left(\sum_{l=1}^{d^{\left(k\right)}}\left(\frac{d}{dt}r_{jl}^{\left(k\right)}\left(t\right)\right)r_{jl}^{(k)*}\left(t\right)\right)|b_{j}^{\left(k\right)}\rangle \langle b_{j}^{\left(k\right)}|,$$
and
$${\hat{H}}_{X}^{\left(k\right)}\left(t\right)\equiv i\hbar \sum_{j=1}^{d^{\left(k\right)}}\sum_{m\neq j}^{d^{\left(k\right)}}\left(\sum_{l=1}^{d^{\left(k\right)}}\frac{d}{dt}r_{jl}^{\left(k\right)}\left(t\right)r_{ml}^{(k)*}\left(t\right)\right)e^{\frac{i}{\hbar }\left(b_{m}^{\left(k\right)}-b_{j}^{\left(k\right)}\right)t}|b_{j}^{\left(k\right)}\rangle \langle b_{m}^{\left(k\right)}|.$$

### 4.2. Local Internal Energy Additivity

Given Equation (Equation 2) and (Equation 3) for the Schmidt decomposition and the Schmidt basis dynamics, we have the following equation
$$i\hbar \frac{d}{dt}|\Psi \left(t\right)\rangle =\left({\tilde{H}}^{\left(1\right)}\left(t\right)+{\tilde{H}}^{\left(2\right)}\left(t\right)\right)|\Psi \left(t\right)\rangle +\sum_{j=1}^{d^{\left(1\right)}}\left(i\hbar \frac{d}{dt}\lambda_{j}\left(t\right)\right)|\varphi_{j}^{\left(1\right)}\left(t\right)\rangle ⊗|\varphi_{j}^{\left(2\right)}\left(t\right)\rangle .$$
Thus, since $i\hbar \frac{d}{dt}|\Psi \left(t\right)\rangle ={\hat{H}}^{\left(0\right)}|\Psi \left(t\right)\rangle$, we obtain
$$\begin{array}{c} \langle \Psi \left(t\right)|{\hat{H}}^{\left(0\right)}|\Psi \left(t\right)\rangle =\langle \Psi \left(t\right)|\left({\tilde{H}}^{\left(1\right)}\left(t\right)+{\tilde{H}}^{\left(2\right)}\left(t\right)\right)|\Psi \left(t\right)\rangle +\;\;\;\;\;\;\;\;\;\;\;\;\;\;\;\;\;\;\;\;\;\\ \langle \Psi \left(t\right)|\sum_{j=1}^{d^{\left(1\right)}}\left(i\hbar \frac{d}{dt}\lambda_{j}\left(t\right)\right)|\varphi_{j}^{\left(1\right)}\left(t\right)\rangle ⊗|\varphi_{j}^{\left(2\right)}\left(t\right)\rangle . \end{array}$$
Nevertheless, notice that due to the normalization of |Ψ(t)〉, the second contribution is necessarily null. Therefore,
(14)$$\langle {\hat{H}}^{\left(0\right)}\rangle =\langle {\tilde{H}}^{\left(1\right)}\left(t\right)\rangle +\langle {\tilde{H}}^{\left(2\right)}\left(t\right)\rangle =U^{\left(0\right)}.$$

### 4.3. Generalization for Mixed States

The generalization of our formalism for mixed states is straightforward. If the universe’s quantum state is described by the following ensemble of pure states ${\hat{\rho }}^{\left(0\right)}\left(t\right)\equiv \sum_{\eta =1}^{d^{\left(0\right)}}P_{\eta }|\Psi_{\eta }\left(t\right)\rangle \langle \Psi_{\eta }\left(t\right)|$, where $\langle \Psi_{\alpha }\left(t\right)|\Psi_{\beta }\left(t\right)\rangle =\delta_{\alpha \beta }$ and $\sum_{\eta =1}^{d^{\left(0\right)}}P_{\eta }=1$, then the previous procedure can be applied to every individual element of the ensemble: for every ket $|\Psi_{\eta }\left(t\right)\rangle$, we have an associate Schmidt decomposition
(15)$$|\Psi_{\eta }\left(t\right)\rangle =\sum_{j=1}^{d^{\left(1\right)}}\lambda_{\eta j}\left(t\right)|\varphi_{\eta j}^{\left(1\right)}\left(t\right)\rangle ⊗|\varphi_{\eta j}^{\left(2\right)}\left(t\right)\rangle ,$$
where $\lambda_{\eta j}\left(t\right)$ and $|\varphi_{\eta j}^{\left(1,2\right)}\left(t\right)\rangle$ are its respective Schmidt coefficients and basis; thus, we can define the ηth local effective Hamiltonian ${\tilde{H}}_{\eta }^{\left(k\right)}\left(t\right)\equiv i\hbar \sum_{j=1}^{d^{\left(k\right)}}(\frac{d}{dt}|\varphi_{\eta j}^{\left(k\right)}\left(t\right)\rangle )\langle \varphi_{\eta j}^{\left(k\right)}\left(t\right)|$ for subsystem (k), such that ${\tilde{H}}_{\eta }^{\left(k\right)}\left(t\right)={\hat{H}}^{\left(k\right)}+{\hat{H}}_{LS;\eta }^{\left(k\right)}\left(t\right)+{\hat{H}}_{X;\eta }^{\left(k\right)}\left(t\right)$.

In this case, the total internal energy $U^{\left(0\right)}\equiv \langle {\hat{H}}^{\left(0\right)}\rangle$ is given by the following average over the ensemble of pure states $U^{\left(0\right)}=\sum_{\eta =1}^{d^{\left(0\right)}}P_{\eta }U_{\eta }^{\left(0\right)}$, where $U_{\eta }^{\left(0\right)}\equiv tr\{{\hat{H}}^{\left(0\right)}|\Psi_{\eta }\left(t\right)\rangle \langle \Psi_{\eta }\left(t\right)|\}$ is immediately recognized as the universe’s internal energy relative to the ηth state $|\Psi_{\eta }\left(t\right)\rangle$, which also satisfies $\frac{d}{dt}U_{\eta }^{\left(0\right)}=0$ for all η. As expected, we can also show that every energy element will satisfy the bipartition additivity property, such that
(16)$$U_{\eta }^{\left(0\right)}=U_{\eta }^{\left(1\right)}\left(t\right)+U_{\eta }^{\left(2\right)}\left(t\right),$$
where $U_{\eta }^{\left(k\right)}\left(t\right)\equiv \langle \Psi_{\eta }\left(t\right)|{\tilde{H}}_{\eta }^{\left(k\right)}\left(t\right)|\Psi_{\eta }\left(t\right)\rangle$. However, the entire subsystem’s internal energy is simply identified as the averages of its possible values, i.e., $U^{\left(k\right)}\left(t\right)=\sum_{\eta =1}^{d^{\left(0\right)}}P_{\eta }U_{\eta }^{\left(k\right)}$ such that
(17)$$U^{\left(0\right)}=U^{\left(1\right)}\left(t\right)+U^{\left(2\right)}\left(t\right).$$

Additionally, it is worth mentioning that this description allows us to represent and characterize systems initially prepared at thermal states with the inverse of temperature β, i.e., if the universe’s Hamiltonian is given by ${\hat{H}}^{\left(0\right)}\equiv \sum_{\eta =1}^{d^{\left(0\right)}}b_{\eta }^{\left(0\right)}|b_{\eta }^{\left(0\right)}\rangle \langle b_{\eta }^{\left(0\right)}|$ then $P_{\eta }\equiv e^{-\beta b_{\eta }^{\left(0\right)}}/Z^{\left(0\right)}$, with $Z^{\left(0\right)}=tr\left\{e^{-\beta b_{\eta }^{\left(0\right)}}\right\}$ and $|\Psi_{\eta }\left(t_{0}\right)\rangle \equiv |b_{\eta }^{\left(0\right)}\rangle$, such that ${\hat{\rho }}^{\left(0\right)}\left(t\right)\equiv \sum_{\eta =1}^{d^{\left(0\right)}}e^{-\beta b_{\eta }^{\left(0\right)}}/Z^{\left(0\right)}|b_{\eta }^{\left(0\right)}\rangle \langle b_{\eta }^{\left(0\right)}|={\hat{\rho }}_{th}^{\left(0\right)}$. Note that, despite the universe state being thermal, the local states do not necessarily preserve the Gibbs form relative to their bare Hamiltonians, especially under strong coupling regimes. In fact, they are given by the following general and exact expression
(18)$${\hat{\rho }}^{\left(k\right)}\left(t\right)=\sum_{\eta =1}^{d^{\left(0\right)}}\frac{e^{-\beta b_{\eta }^{\left(0\right)}}}{Z^{\left(0\right)}}\sum_{j=1}^{d^{\left(1\right)}}\lambda_{\eta j}^{2}\left(t\right)|\varphi_{\eta j}^{\left(k\right)}\left(t\right)\rangle \langle \varphi_{\eta j}^{\left(k\right)}\left(t\right)|.$$

In the case where the whole universe is considered to be in a thermal state characterized by the inverse temperature β, the so-called Hamiltonian of the mean force [46,47] of subsystem (1)
(19)$${\hat{H}}_{MF}^{\left(1\right)}\equiv -\frac{1}{\beta }ln\left(\frac{tr_{2}\left\{e^{-\beta {\hat{H}}^{\left(0\right)}}\right\}}{tr_{2}\left\{e^{-\beta {\hat{H}}^{\left(2\right)}}\right\}}\right)$$
is usually defined and commonly understood as its effective Hamiltonian in a way that the local equilibrium state becomes an effective Gibbs one: ${\hat{\rho }}^{\left(1\right)}=tr_{2}\left\{{\hat{\rho }}_{th}^{\left(0\right)}\right\}\equiv e^{-\beta {\hat{H}}_{MF}^{\left(1\right)}}/Z_{MF}^{\left(1\right)}$, where $Z_{MF}^{\left(1\right)}\equiv tr_{1}\left\{e^{-\beta {\hat{H}}_{MF}^{\left(1\right)}}\right\}$ [37,48,49]. Such an operator naturally encompasses both the temperature and the interaction term. More importantly, this formulation suggests that the local internal energy of the subsystem in question may simply be obtained by $U_{MF}^{\left(1\right)}\equiv -\partial_{\beta }ln\left(Z_{MF}^{\left(1\right)}\right)$. Alternatively, one can define the effective energy operator ${\hat{E}}_{MF}^{\left(1\right)}=\partial_{\beta }\left(\beta {\hat{H}}_{MF}^{\left(1\right)}\right)$ and express $U_{MF}^{\left(1\right)}$ as the following expectation value $U_{MF}^{\left(1\right)}=tr_{1}\left\{{\hat{\rho }}^{\left(1\right)}{\hat{E}}_{MF}^{\left(1\right)}\right\}$. It is important to mention that, in general, the operators ${\hat{H}}^{\left(1\right)}$, ${\hat{H}}_{MF}^{\left(1\right)}$ and ${\hat{E}}_{MF}^{\left(1\right)}$ are not necessarily equivalent. Under these circumstances, it would be interesting to understand: (i) the eventual relationships between the local effective Hamiltonians ${\tilde{H}}_{\eta }^{\left(1\right)}\left(t\right)$ introduced here and the operators shown above; (ii) the implications of the different definitions of local energies $U^{\left(1\right)}\left(t\right)$ and $U_{MF}^{\left(1\right)}$. In this sense, it is interesting to notice that while $U^{\left(1\right)}\left(t\right)$ is directly related to ${\tilde{H}}_{\eta }^{\left(1\right)}\left(t\right)$, $U_{MF}^{\left(1\right)}$ depends on ${\hat{H}}_{MF}^{\left(1\right)}$ and its derivative. Additionally, it is important to emphasize that the additive property is not observed for ${\hat{U}}_{MF}^{\left(1,2\right)}$ in general cases, as will be seen in the example below. Thus, the energy flowing from subsystem (1) is not entirely acquired by subsystem (2), i.e., it is required to include an additional energetic source or sink to guarantee the universe’s energy balance. Nevertheless, a priori, it is unclear how to establish a general and formal connection between all these quantities. In this wise, future research will be necessary to elucidate these questions in depth.

However, despite the similar strategy of defining an effective Hamiltonian and effective local energies, the framework presented above lacks generality, since it is restricted to a particular scenario of thermal equilibrium. In this sense, any other situation is outside its scope. Additionally, once considering the entire bipartition, the energy additivity property is not required or expected to be satisfied. These crucial drawbacks do not happen in the proposal introduced in this paper.

### 4.4. Example

Finally, we illustrate the concepts presented in this paper with a simple–but paradigmatic–example as a proof of principle. Let us consider as our universe two distinguishable interacting two-level systems described by the following symmetrical Hamiltonian:(20)$${\hat{H}}^{\left(0\right)}=\hbar \frac{\omega }{2}{\hat{\sigma }}_{z}^{\left(1\right)}⊗{\hat{1}}^{\left(2\right)}+{\hat{1}}^{\left(1\right)}⊗\hbar \frac{\omega }{2}{\hat{\sigma }}_{z}^{\left(2\right)}+\hbar g{\hat{\sigma }}_{z}^{\left(1\right)}⊗{\hat{\sigma }}_{z}^{\left(2\right)}$$
where ${\hat{H}}^{\left(1,2\right)}\equiv \hbar \frac{\omega }{2}{\hat{\sigma }}_{z}^{\left(1,2\right)}$ are the local bare Hamiltonians, ${\hat{H}}_{int}\equiv \hbar g{\hat{\sigma }}_{z}^{\left(1\right)}⊗{\hat{\sigma }}_{z}^{\left(2\right)}$ is the interaction term, ${\hat{\sigma }}_{z}^{\left(1,2\right)}$ is the usual Pauli matrix such that ${\hat{\sigma }}_{z}^{\left(1,2\right)}|\pm^{\left(1,2\right)}\rangle =\pm |\pm^{\left(1,2\right)}\rangle$, *g* is the coupling constant, and ℏω is the spectral gap for ${\hat{H}}^{\left(1,2\right)}$. Additionally, let us further assume the universe’s state is initially described by the Gibbs ensemble with the inverse of temperature β, i.e., ${\hat{\rho }}^{\left(0\right)}\left(t_{0}\right)=\sum_{\eta =1}^{4}P_{\eta }|\Psi_{\eta }\left(t_{0}\right)\rangle \langle \Psi_{\eta }\left(t_{0}\right)|\equiv e^{-\beta {\hat{H}}^{\left(0\right)}}/Z^{\left(0\right)}$ with $Z^{\left(0\right)}=tr\left\{e^{-\beta {\hat{H}}^{\left(0\right)}}\right\}$.Thus, the set of time-evolved orthonormal pure states is already in the Schmidt decomposition form (see Equation (Equation 15)) and is simply identified as
$$\begin{array}{cc} |\Psi_{1}\left(t\right)\rangle & =e^{-\frac{i}{\hbar }\hbar \left(g+\omega \right)\left(t-t_{0}\right)}|{+}^{\left(1\right)}\rangle ⊗|{+}^{\left(2\right)}\rangle ,\\ |\Psi_{2}\left(t\right)\rangle & =e^{-\frac{i}{\hbar }\hbar \frac{\omega }{2}\left(t-t_{0}\right)}e^{\frac{i}{\hbar }\hbar \frac{\omega }{2}\left(t-t_{0}\right)}e^{\frac{i}{\hbar }\hbar g\left(t-t_{0}\right)}|{+}^{\left(1\right)}\rangle ⊗|{-}^{\left(2\right)}\rangle ,\\ |\Psi_{3}\left(t\right)\rangle & =e^{\frac{i}{\hbar }\hbar \frac{\omega }{2}\left(t-t_{0}\right)}e^{-\frac{i}{\hbar }\hbar \frac{\omega }{2}\left(t-t_{0}\right)}e^{\frac{i}{\hbar }\hbar g\left(t-t_{0}\right)}|{-}^{\left(1\right)}\rangle ⊗|{+}^{\left(2\right)}\rangle ,\\ |\Psi_{4}\left(t\right)\rangle & =e^{-\frac{i}{\hbar }\hbar \left(g-\omega \right)\left(t-t_{0}\right)}|{-}^{\left(1\right)}\rangle ⊗|{-}^{\left(2\right)}\rangle \end{array}$$
Notice that, for every $|\Psi_{\eta }\left(t\right)\rangle$ with η=1,…,4, the Schmidt rank is one, which implies that only a single element of the Schmidt basis for each two-level system, constituted by $\{|\varphi_{\eta +}^{\left(k\right)}\left(t\right)\rangle ,|\varphi_{\eta -}^{\left(k\right)}\left(t\right)\rangle \}$, plays a role in the dynamics of the pure state and, therefore, is relevant for accounting the energetics. Additionally, it is clear that these kets can be written as $|\varphi_{\eta \pm }^{\left(k\right)}\left(t\right)\rangle =e^{-\frac{i}{\hbar }c_{\eta \pm }^{\left(k\right)}\left(t-t_{0}\right)}|\pm^{\left(k\right)}\rangle$, such that $i\hbar \frac{d}{dt}|\varphi_{\eta \pm }^{\left(k\right)}\left(t\right)\rangle =c_{\eta \pm }^{\left(k\right)}|\varphi_{\eta \pm }^{\left(k\right)}\left(t\right)\rangle$ and $c_{\eta \pm }^{\left(k\right)}$ are constant phase coefficients satisfying
$$\begin{array}{cc} c_{1+}^{\left(1\right)}+c_{1+}^{\left(2\right)} & =\hbar \left(g+\omega \right)=U_{1}^{\left(0\right)},\\ c_{2+}^{\left(1\right)}+c_{2-}^{\left(2\right)} & =-\hbar g+\hbar \frac{\omega }{2}-\hbar \frac{\omega }{2}=U_{2}^{\left(0\right)},\\ c_{3-}^{\left(1\right)}+c_{3+}^{\left(2\right)} & =-\hbar g+\hbar \frac{\omega }{2}-\hbar \frac{\omega }{2}=U_{3}^{\left(0\right)},\\ c_{4-}^{\left(1\right)}+c_{4-}^{\left(2\right)} & =\hbar \left(g-\omega \right)=U_{4}^{\left(0\right)}, \end{array}$$
where $U_{\eta }^{\left(0\right)}\equiv \langle \Psi_{\eta }\left(t\right)|{\hat{H}}^{\left(0\right)}|\Psi_{\eta }\left(t\right)\rangle$ is the universe’s internal energy relative to the ηth pure state, and the set of coefficients $\left\{c_{1-}^{\left(1\right)},c_{1-}^{\left(2\right)},c_{2-}^{\left(1\right)},c_{2+}^{\left(2\right)},c_{3+}^{\left(1\right)},c_{3-}^{\left(2\right)},c_{4+}^{\left(1\right)},c_{4+}^{\left(2\right)}\right\}$ is relative to the Schmidt basis elements absent in the pure states, or, analogously, in the local density matrices. Given the nature of this particular Hamiltonian, it is also clear that the phases obtained by the Schmidt basis above during the time-evolution contain both the contribution of the local bare Hamiltonian and the interaction term. Additionally, considering the symmetry of the subsystems, this readily suggests the following identification: $c_{1+}^{\left(1\right)}=c_{1+}^{\left(2\right)}=\hbar \left(g+\omega \right)/2$, $c_{4-}^{\left(1\right)}=c_{4-}^{\left(2\right)}=\hbar \left(g-\omega \right)/2$, $c_{2+}^{\left(1\right)}=c_{3+}^{\left(2\right)}=-\hbar \left(g-\omega \right)/2$ and $c_{2-}^{\left(2\right)}=c_{3-}^{\left(1\right)}=-\hbar \left(g+\omega \right)/2$. As expected, once considering g→0, these phases are exactly the dynamical ones that would be obtained in a free evolution.

Under these circumstances, one can show that the local effective Hamiltonians have the following simple form ${\tilde{H}}_{\eta }^{\left(k\right)}=c_{\eta +}^{\left(k\right)}|{+}^{\left(k\right)}\rangle \langle {+}^{\left(k\right)}|+c_{\eta -}^{\left(k\right)}|{-}^{\left(k\right)}\rangle \langle {-}^{\left(k\right)}|$. While half of the coefficients are already identified, the remaining ones can be determined as long we guarantee that ${\tilde{H}}_{\eta }^{\left(k\right)}\to {\hat{H}}^{\left(k\right)}$ whenever g→0. Thus, one can write
$$\begin{array}{cc} {\tilde{H}}_{1}^{\left(1\right)} & =\frac{\hbar }{2}\left(\omega +g\right){\hat{\sigma }}_{z}^{\left(1\right)},\;{\tilde{H}}_{1}^{\left(2\right)}=\frac{\hbar }{2}\left(\omega +g\right){\hat{\sigma }}_{z}^{\left(2\right)}\\ {\tilde{H}}_{2}^{\left(1\right)} & =\frac{\hbar }{2}\left(\omega -g\right){\hat{\sigma }}_{z}^{\left(1\right)},\;{\tilde{H}}_{2}^{\left(2\right)}=\frac{\hbar }{2}\left(\omega +g\right){\hat{\sigma }}_{z}^{\left(2\right)}\\ {\tilde{H}}_{3}^{\left(1\right)} & =\frac{\hbar }{2}\left(\omega +g\right){\hat{\sigma }}_{z}^{\left(1\right)},\;{\tilde{H}}_{3}^{\left(2\right)}=\frac{\hbar }{2}\left(\omega -g\right){\hat{\sigma }}_{z}^{\left(2\right)}\\ {\tilde{H}}_{4}^{\left(1\right)} & =\frac{\hbar }{2}\left(\omega -g\right){\hat{\sigma }}_{z}^{\left(1\right)},\;{\tilde{H}}_{4}^{\left(2\right)}=\frac{\hbar }{2}\left(\omega -g\right){\hat{\sigma }}_{z}^{\left(2\right)} \end{array}$$
such that the local internal energies, $U_{\eta }^{\left(k\right)}\equiv \langle \Psi_{\eta }\left(t\right)|{\tilde{H}}_{\eta }^{\left(k\right)}|\Psi_{\eta }\left(t\right)\rangle$, are identified by
$$\begin{array}{cc} U_{1}^{\left(1\right)} & =c_{1+}^{\left(1\right)}=U_{1}^{\left(2\right)}=\frac{\hbar }{2}\left(\omega +g\right),\\ U_{2}^{\left(1\right)} & =c_{2+}^{\left(1\right)}=U_{3}^{\left(2\right)}=\frac{\hbar }{2}\left(\omega -g\right),\\ U_{2}^{\left(2\right)} & =c_{2-}^{\left(2\right)}=U_{3}^{\left(1\right)}=-\frac{\hbar }{2}\left(\omega +g\right),\\ U_{4}^{\left(1\right)} & =c_{4-}^{\left(1\right)}=U_{4}^{\left(2\right)}=-\frac{\hbar }{2}\left(\omega -g\right). \end{array}$$
As expected, it is clear that the additivity is fulfilled for all η, i.e., $U_{\eta }^{\left(0\right)}=U_{\eta }^{\left(1\right)}+U_{\eta }^{\left(2\right)}$. Additionally, this property remains satisfied from the perspective of the whole ensemble of pure states since
(21)$$U^{\left(0\right)}=\sum_{\eta =1}^{4}P_{\eta }U_{\eta }^{\left(0\right)}=U^{\left(1\right)}+U^{\left(2\right)},$$
where $U^{\left(0\right)}$ is the total internal energy of the universe and $U^{\left(k\right)}=\sum_{\eta =1}^{4}P_{\eta }U_{\eta }^{\left(k\right)}$ is the whole internal energy attributed to subsystem (k). It is worth mentioning that we might consider different phase gauges by simply adding a constant value in any local effective Hamiltonian of subsystem (1) and consistently subtracting the same quantity in the subsystem (2) counterpart. Naturally, this would imply different absolute values of energy but would preserve both the spectral gaps and the additivity property.

Additionally, it is interesting to compute the Hamiltonian of the mean force, Equation (Equation 19), relative to this example and its respective effective energy operator ${\hat{E}}_{MF}^{\left(1,2\right)}=\partial_{\beta }\left(\beta {\hat{H}}_{MF}^{\left(1,2\right)}\right)$. Hence, we can show that
$$\begin{array}{cc} {\hat{H}}_{MF}^{\left(1,2\right)}= & -\frac{1}{\beta }\ln\left(\frac{2\cosh\left(\beta \hbar (g+\frac{\omega }{2})\right)}{e^{\beta \omega \hbar }+1}\right)|{+}^{\left(1,2\right)}\rangle \langle {+}^{\left(1,2\right)}|\\ & -\frac{1}{\beta }\ln\left(\left(\tanh\left(\frac{\beta \omega \hbar }{2}\right)+1\right)\cosh\left(\beta \hbar (g-\frac{\omega }{2})\right)\right)|{-}^{\left(1,2\right)}\rangle \langle {-}^{\left(1,2\right)}|\\ {\hat{E}}_{MF}^{\left(1,2\right)}= & \left(\frac{\omega \hbar e^{\beta \omega \hbar }}{e^{\beta \omega \hbar }+1}-\frac{\hbar }{2}\left(2g+\omega \right)\tanh\left(\frac{\beta \hbar }{2}\left(2g+\omega \right)\right)\right)|{+}^{\left(1,2\right)}\rangle \langle {+}^{\left(1,2\right)}|\\ \; & +\frac{\hbar }{2}\left(\left(\omega -2g\right)\tanh\left(\beta g\hbar -\frac{\beta \omega \hbar }{2}\right)-\frac{2\omega }{e^{\beta \omega \hbar }+1}\right)|{-}^{\left(1,2\right)}\rangle \langle {-}^{\left(1,2\right)}|. \end{array}$$

Thus, it is clear that ${\hat{H}}_{MF}^{\left(1,2\right)}\neq {\hat{E}}_{MF}^{\left(1,2\right)}\neq {\tilde{H}}_{\eta }^{\left(1,2\right)}$ and $\left[{\hat{H}}_{MF}^{\left(1,2\right)},{\tilde{H}}_{\eta }^{\left(1,2\right)}\right]=\left[{\hat{E}}_{MF}^{\left(1,2\right)},{\tilde{H}}_{\eta }^{\left(1,2\right)}\right]=0$ for all η. Finally, the local internal energies obtained from this framework are given by
$$\begin{array}{c} \begin{array}{cc} & U_{MF}^{\left(1,2\right)}=\frac{\hbar \left(2g\left(e^{\beta \omega \hbar }+1\right)\left(e^{2\beta \omega \hbar }-2e^{\beta \hbar (2g+\omega )}+1\right)\right)}{2\left(e^{\beta \omega \hbar }+1\right)\left(e^{2\beta \omega \hbar }+2e^{\beta \hbar (2g+\omega )}+1\right)}\\ & \;\;\;\;\;\;\;\;\;\;\;\;\;\;\;\;-\frac{\hbar \omega \left(e^{\beta \omega \hbar }-1\right)\left(e^{\beta \omega \hbar }\left(e^{\beta \omega \hbar }-2e^{2\beta g\hbar }+4\right)+1\right)}{2\left(e^{\beta \omega \hbar }+1\right)\left(e^{2\beta \omega \hbar }+2e^{\beta \hbar (2g+\omega )}+1\right)}\\ & \;\;\;\;\;\;\;\;\;\;⟹U_{MF}^{\left(1\right)}+U_{MF}^{\left(2\right)}\neq U^{\left(0\right)} \end{array} \end{array}$$
Notice that, despite also providing an effective description of the local internal energies, in the sense that it also contains the contribution of the interaction term, the quantities $U_{MF}^{\left(1,2\right)}$ and $U^{\left(1,2\right)}$ do not agree with each other. More importantly, the additivity of energy is not satisfied, i.e., $U_{MF}^{\left(1\right)}+U_{MF}^{\left(2\right)}\neq U^{\left(0\right)}=U^{\left(1\right)}+U^{\left(2\right)}$.

References and Note

## Data Availability

Not applicable.

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
