# Peer review of "A Schmidt Decomposition Approach to Quantum Thermodynamics"

_entropy, 2022, doi:10.3390/e24111645_

Round 1
Reviewer 1 Report
In the present work, the authors tried to developed a self-consistent approach for identifying local effective Hamiltonians which is quite desirable at strong couplings. The framework is nice and the presentation is very clear. I just have one comment: Could the authors discuss the relation (i.e., similarity and difference) between the so-obtained local effective Hamiltonian and the existing concept of Hamiltonian of mean force when considering a single thermal bath?
Author Response
Point 1: I just have one comment: Could the authors discuss the relation (i.e., similarity and difference) between the so-obtained local effective Hamiltonian and the existing concept of Hamiltonian of mean force when considering a single thermal bath?
Response 1: Thank you for your suggestion and review. Even though we considered pure states as more appropriate for our initial presentation, we believe that you provided valuable input. In order to briefly discuss it, we have included some important sections in the “Supplementary Materials”. In particular, we both added a generalization of our procedure for mixed states, which was crucial to consider global Gibbs states, and introduced the concept of the Hamiltonian of mean force (HMF) and the energy operator associated with it. We also added a very simple paradigmatic example as proof of principle, which provides the means to illustrate our procedure and their differences. More specifically, we were able to verify that the additivity property is not guaranteed for the internal energies derived from the HFM, which is a fundamental difference. Nevertheless, we also mentioned that the eventual formal relationship between the local effective Hamiltonians and the HMF is not straightforward and highlighted that more research is necessary to elucidate these questions in depth. Finally, it is worth mentioning that we believe that your input positively contributed to improving our manuscript.
The colour used to highlight these points is “teal”.
Reviewer 2 Report
This paper proposes a definition of quantum systems' local internal (thermodynamic) energy. Consider, for instance, two strongly coupled subsystems. Using the Schmidt decomposition for the correlated quantum state, the authors propose a local operator (in fact, one for each subsystem) whose expected value should be considered as the internal energy of the subsystem. The choice is motivated by two premises:
For strongly interacting systems, the local internal energies should be (i) obtained by local measurements and (ii) an additive quantity, i.e., the total energy is the sum of the energies of the subsystems.
Although this additivity may be desirable, the definition is not physically motivated and may hide the physical picture associated with energy. The arbitrariness related to the phase freedom in the Schmidt coefficients brings a new complication that is absent if one is not attached to ideas (i) and (ii).
To convince of the value of their proposition, the authors should include a simple example, for instance, two two-level strongly coupled systems. If the global system is in the Gibbs equilibrium state, the hamiltonian of mean force is usually the operator associated with the internal energy of the subsystem. They could compare them and discuss the advantages of one respect to another and the role of phase freedom in a concrete example.
Author Response
Point 1: Although this additivity may be desirable, the definition is not physically motivated and may hide the physical picture associated with energy. The arbitrariness related to the phase freedom in the Schmidt coefficients brings a new complication that is absent if one is not attached to ideas (i) and (ii).
To convince of the value of their proposition, the authors should include a simple example, for instance, two two-level strongly coupled systems. If the global system is in the Gibbs equilibrium state, the hamiltonian of mean force is usually the operator associated with the internal energy of the subsystem. They could compare them and discuss the advantages of one respect to another and the role of phase freedom in a concrete example.
Response 1: Thank you for your suggestion and review. We strongly agreed that a simple example would positively contribute to our presentation. In order to do that, we have included some important sections in the “Supplementary Materials”: first, we introduced a generalization of our procedure for mixed states, which was necessary to consider global Gibbs states; then, we also briefly presented the concept of the Hamiltonian of mean force (HMF) and the related quantities for accounting for internal energy; finally, we presented a simple paradigmatic example as proof of principle to illustrate the procedure and elucidate the differences between our proposal and the HMF.
Along these lines, we accepted your suggestion, such that our example consists of two two-level interacting systems described by a thermal global state. On the one hand, we were able to show that the additivity property is not guaranteed for the internal energies derived from the HFM, which is a crucial difference. On the other hand, we also mentioned that the eventual formal relationship between the local effective Hamiltonians and the HMF is not straightforward. For this reason, we highlighted that further research is necessary to elucidate these questions in depth. Finally, we deeply appreciate your input, and we also believe that this strengthened our manuscript.
The colour used to highlight these points is “brown”.
Reviewer 3 Report
The paper is devoted to a proposal of local Hamiltonians describing the energy of subsystems for the case of a bipartite system in the presence of an interaction term. The overall system is supposed to start in a pure state.
The idea of the authors it to consider the instantaneous Schmidt decomposition of the evolved state in order to infer a local operator determining the evolution of the local Schmidt bases.
In such a way they obtain two local effective Hamiltonians that by construction recover the total conserved energy. The authors insist that these two features justify the introduction of these effective operators.
They further analyze the ambiguities inherent in the proposed definition.
I find the contribution can be of interest to the readership of Entropy, but I think the author should clarify a few points.
The whole formalism is based on taking an initial pure state. This assumption does not appear very realistic in situations relevant for quantum thermodynamics so that the authors should comment on this point. They speak about a quantum universe, but it would be useful to connect to possible relevant situations.
If I understand correctly, In essence the authors find an implicit way to distribute the energy due to the interaction term between the two interacting systems in a time-dependent way. It would be interesting if the authors could give some insight into how this term is distributed among the two effective Hamiltonians.
The authors speak about the additivity of energy, but of course, this construction is restricted to two systems, given that it builds on the Schmidt decomposition. The authors should clarify that this should not be confused with the general notion of additivity of energy.
In the introduction the sentence "some open questions concerning central
aspects of the theory are exceptionally notorious" is not clear.
Author Response
Point 1: The whole formalism is based on taking an initial pure state. This assumption does not appear very realistic in situations relevant for quantum thermodynamics so that the authors should comment on this point. They speak about a quantum universe, but it would be useful to connect to possible relevant situations.
Response 1: First, we would like to thank you for your suggestions and review. Even though we considered pure states as more appropriate for our first presentation, we believe that you raised a very important question.
In order to cover this aspect, we have included some major sections in the “Supplementary Materials”. In particular, we introduced the generalization of our procedure to include mixed states, which is a fundamental step to considering thermal ones, for instance. Besides, as proof of principle, we also included a simple paradigmatic example to illustrate our procedure and elucidate its validity for relevant situations.
Along these lines, we strongly believe that your suggestion positively contributed to improving our manuscript.
The colour used to highlight these changes is “magenta”.
Point 2: If I understand correctly, In essence the authors find an implicit way to distribute the energy due to the interaction term between the two interacting systems in a time-dependent way. It would be interesting if the authors could give some insight into how this term is distributed among the two effective Hamiltonians.
Response 2: Exactly, you are completely right. Along these lines, we are able to connect the expectation values of the interaction term with the time-dependent operators H_{LS}(t) and H_{X}(t) of both subsystems. In fact, we can show that the interaction energy is symmetrically decomposed into these local expectation values. However, it is worth mentioning that it does not mean that their numerical values cannot be extremely different. We believe that the addition of this expression is indeed valuable for our presentation, and it was included in a paragraph in the "Internal energies and additivity" section.
Point 3: The authors speak about the additivity of energy, but of course, this construction is restricted to two systems, given that it builds on the Schmidt decomposition. The authors should clarify that this should not be confused with the general notion of additivity of energy.
Response 3: Considering this, some minor additions were performed to the text. Hopefully, it is clearer now.
Point 4: In the introduction the sentence "some open questions concerning central aspects of the theory are exceptionally notorious" is not clear.
Response 4: Thank you for pointing this out. We restructured this phrase and its connection with the following ones. Hopefully, we expect we managed to improve our main message, and you can find it more clear now.
Reviewer 4 Report
I believe the present manuscript makes a timely and significant contribution to quantum thermodynamics towards a unifying formalism thorough Schmidt decomposition.
I can say that as a researcher with quantum information background and conducting research in quantum thermodynamics, this manuscript has a great potential to help answering fundamental questions we have been asking so far. It is solid and very well written. Its mathematics is easy to follow.
Adapting the manuscripts format to MDPI would require re-organization of the sections and presentation of notation, etc.
It is also rising interesting open questions. One is about bound entanglement in thermodynamics; please see
Are the Laws of Entanglement Theory Thermodynamical? Phys. Rev. Lett. 89, 240403 (2002). https://doi.org/10.1103/PhysRevLett.89.240403
Work and heat value of bound entanglement. Quantum Inf Process 18, 373 (2019). https://doi.org/10.1007/s11128-019-2488-y
Because Schmidt-numbers can witness bound entanglement, please see
Schmidt-number witnesses and bound entanglement, Phys. Rev. A 63, 050301(R). https://doi.org/10.1103/PhysRevA.63.050301
I think this might be interesting for future research, and I recommend authors to add a discussion about it.
Also, Because heat engines are crucial in quantum thermodynamics, I can recommend authors to improve that part by mentioning more recent works such as
Temperature control in dissipative cavities by entangled dimers, J. Phys. Chem. C 2019, 123, 7. https://doi.org/10.1021/acs.jpcc.8b11445
Work harvesting by q-deformed statistical mutations in an Otto engine, https://doi.org/10.48550/arXiv.2208.08565
Superradiant Quantum Heat Engine. Sci Rep 5, 12953 (2015). https://doi.org/10.1038/srep12953
Author Response
Point 1: It is also rising interesting open questions. One is about bound entanglement in thermodynamics; please see
Are the Laws of Entanglement Theory Thermodynamical? Phys. Rev. Lett. 89, 240403 (2002). https://doi.org/10.1103/PhysRevLett.89.240403
Work and heat value of bound entanglement. Quantum Inf Process 18, 373 (2019). https://doi.org/10.1007/s11128-019-2488-y
Because Schmidt-numbers can witness bound entanglement, please see
Schmidt-number witnesses and bound entanglement, Phys. Rev. A 63, 050301(R). https://doi.org/10.1103/PhysRevA.63.050301
I think this might be interesting for future research, and I recommend authors to add a discussion about it.
Response 1: Thank you for your valuable suggestion, review and references. Indeed, we agree that this is a very interesting topic for future research that we did not think of before. We are grateful that you point this out to us.
We included a brief paragraph in the "Conclusion" section mentioning this topic and indicating it as a compelling direction for future work. We deeply appreciate your input, and we also believe that this strengthened our manuscript.
The colour used to highlight these changes is “blue”.
Point 2: Also, Because heat engines are crucial in quantum thermodynamics, I can recommend authors to improve that part by mentioning more recent works such as
Temperature control in dissipative cavities by entangled dimers, J. Phys. Chem. C 2019, 123, 7. https://doi.org/10.1021/acs.jpcc.8b11445
Work harvesting by q-deformed statistical mutations in an Otto engine, https://doi.org/10.48550/arXiv.2208.08565
Superradiant Quantum Heat Engine. Sci Rep 5, 12953 (2015). https://doi.org/10.1038/srep12953
Response 2: These are some interesting references that we missed. Thank you for calling our attention and indicating them to us.
We agree entirely that heat engines are one of the most critical topics in the field nowadays, so these papers should be mentioned in our references.
We included them in the "Introduction" section, which definitely represents an improvement to our manuscript.
Round 2
Reviewer 2 Report
The authors have addressed all the points raised by the referees.
I do not have further comments, and I think the paper should be published.